# Profitability determinants of the natural stone industry: Evidence from Spain and Italy

**Fernando José Zambrano Farías** [1,2] *, **María del Carmen Valls Martínez**[3], **Pedro Antonio Martín-Cervantes**[3]

1 Faculty of Administrative Sciences, International University of Ecuador, Quito, Ecuador, 2 University of Guayaquil, Guayaquil, Ecuador, 3 Mediterranean Research Center on Economics and Sustainable Development, Economics and Business Department, University of Almería, Almería, Spain

☯ These authors contributed equally to this work.
* fezambranofa@uide.edu.ec

**Data Availability Statement:** The data are third party data extracted from Amadeus data base. You may access the database at the following link:

## Abstract

The natural stone sector is an important driver of the Spanish and Italian economies, which underwent internationalization after the financial crisis of 2008 as part of a survival and development strategy. This article aims to study the financial and economic profitability of this sector in the two leading European production countries, as well as its determinants. For this purpose, the economic-financial data of a sample composed of 453 companies (203 Spanish and 250 Italian) from 2015–2019 were analyzed using the multiple linear regression methodology. To address the problems of possible endogeneity and omission of variables in the model, the dependent variable was used as a regressor with one and two lags, and panel data with fixed effects were considered after performing the Hausman test. The results show significant differences between the two countries, with higher profitability in Italy. Company size, company growth (measured as the change in assets), and the variation in the country's GDP all positively affected profitability. At the same time, the level of indebtedness showed a negative relationship. The country's inflation rate and gender diversity in top management were shown to be non-relevant variables. The research conducted indicates that, to increase profitability, Spanish and Italian companies in the natural stone sector should undergo mergers in order to grow in size, increase efficiency in the use of assets, reduce their dependence on external financing, and promote equity capital. In addition, Italian companies should reduce the average period of payment to suppliers to lower deferral costs, and boost exports to become less dependent on the country's domestic economy.

## 1. Introduction

In Spain and Italy, the natural stone sector is a traditional sector made up of marble, slate, and granite production. The sector's economic situation has been badly hit as a result of the crisis in the construction sector. In 2018, Italy and Spain were the fifth and eighth largest producers of natural stone in the world, respectively. This sector has a very important specific weight within the economy of both countries, which makes it relevant to analyze the profitability of the companies that comprise it.

(https://authenticate.bvdep.com/rediris). To access the database, the user must belong to an organization, usually a university, with access to the database, i.e. he/she must have a validated username and password to access. The search used corresponds to companies with the NACE code 0811(National Classification of Economic Activities: extraction of ornamental and building stone, limestone, gypsum, chalk, and slate) in Spain and Italy, for the years 2015 to 2019, and the variables: Año (year), Id (identifier), Código ISO del país (ISO country code), Fecha de constitución (creation date), Activo total (total asset), Forma jurídica estándar (legal form), Stock, Deudores (debtors),Pasivo total (total liabilities), Pasivos no corrientes (current liabilities) Ingresos explotación (operating income), Resultado ejercicio (net income), Resultado Actividades ordinarias (result ordinary activities), Directores/Administradores (managers). The authors confirm that they have accessed this database as professors of the University to which they belong and which is associated with this database.

**Funding:** This study was supported by the Factores Explicativos de la rentabilidad de las microempresas en Ecuador, International University of Ecuador in the form of a grant to FJZF [UIDE-DGIP-GYE-PROY-20-002] and by the University of Almería: Research group in Ethics, Gender and Sustainability (SEJ-647) in the form of grant to MCVM [PPUENTE2022/006].

**Competing interests:** The authors have declared that no competing interests exist.

In Europe, two producer nations, Spain and Italy, have traditionally stood out. Both countries have underpinned their leadership in the sector over the last decade, with a gradual increase in gross exports, and have carried out ambitious innovation and sustainability processes that have allowed them to find new market niches focused on both the extraction and production of natural stone, as well as on the elaboration of other types of construction materials obtained from the byproducts of the handling and polishing phases (e.g., ceramic products, mortars, and concretes). Spain and Italy have undoubtedly strengthened their positions as exporters in the natural stone market since 2013. The economies of these countries tried to mitigate the financial collapse that occurred during the European sovereign debt crisis through projects focused on internationalization and continuous improvement of the competitiveness of their respective productive fabrics [1]. This context defines the marble and other natural stone sector in both Spain and Italy, which finds itself in an expansionary phase following the financial crisis that affected the European construction sector during the 2008–2009 biennium. This sector, considered traditional in both countries, achieved a volume of revenue in 2017 of almost 3.4 billion euros thanks to the recovery in export volumes, in which unprocessed natural materials stand out, with an increase of 1.6% compared to 2016,. In Spain, as in Italy, the natural stone sector is undergoing a stage of transformation in the international economic panorama.

In both countries, the business structure is mostly comprised of family businesses exploiting enormous potential quarries and having few employees on their payroll. An important aspect is the existence of retail companies that bring together hundreds of small companies supplied by the natural stone processing industry. After the financial crisis of 2008, many of these small companies had to cease their operations. Those that remain today survived thanks to the process of internationalization that both economies underwent.

It is well known that the main objective of companies is to maximize profits while minimizing losses. In order to make profits, a company periodically evaluates its results based on its profitability and compares these results with the initial set targets. The theory states that a company has grown if its profitability has increased [2–4]. In contrast, the shares of a less profitable firm will decrease in value [5–10]. In order to achieve the desired profitability, the company must create a strategic plan. However, whether this profitability can be attained depends on many factors. The literature has classified these factors into three categories: (i) endogenous factors, or factors specific to each company, such as financial ratios, age, and number of employees; (ii) factors associated with the industry to which it belongs, among which stand out the geographical location of the company, the size of the industry, and macroeconomic variables; (iii) factors associated with the management capacity of the owner or shareholders [11, 12].

Among the factors most commonly used by researchers are liquidity, leverage, company growth, size, revenue level, inflation, and the gross domestic product (GDP) of the country in which the company operates [13–20].

Several theories try to explain the relationship between these variables and profitability. The Pecking Order Theory establishes that the company should select financial sources in a hierarchical order, and only move on to the next level once the previous one has been exhausted. The company's own resources will be used first, followed by debt, and finally equity, with an inverse relationship between the level of indebtedness and profitability [21]. According to the Resource-Based View Theory, the volume of assets and their level of growth are directly related to profitability [22]. Other theories that have traditionally been used to explain the relationship between these variables and others, such as gender in management and profitability, are Agency Theory [23], Resource Dependency Theory [24], and Stakeholder Theory [25].

A literature review shows that research results differ depending on the context in which the studies are conducted, and depend on factors such as geographical location, company type,

company size, and sector of activity. With respect to the influence of company size on profitability, the results are mixed. For example, studies conducted on Indonesian and Nigerian manufacturing firms in the periods 2013–2015 [26] and 1999–2007 [27], respectively, found that the relationship is positive. In contrast, a study performed on Norwegian salmon farms in the period 2000–2014 [28] found a negative relationship. However, a study of Malaysian [29] construction companies found no relationship between the two variables. The same is true for the relationship between indebtedness and profitability. Some studies, such as those conducted on US [30] and French [31] companies, reveal a positive relationship. However, other empirical analyses performed on companies belonging to G-7 countries [32] and on Ethiopian companies [33] concluded that the relationship is negative. However, a study of UK companies [34] showed no relationship at all.

Despite the extensive literature, this lack of consensus leads to a need for further research into this topic. The heterogeneity of contexts, as well as of methodologies and variables used in the empirical studies, may lead to mixed results which cannot be used for comparison. So far, no research has focused on the natural stone sector in Spain and Italy, or compared both countries to establish analogies and differences. This study aims to fill this gap in the research and to identify the factors affecting profitability, measured by return on equity (ROE) and return on assets (ROA), through a panel data regression. This empirical analysis provides evidence on the most relevant variables influencing performance [2] and contributes to current knowledge. First, it analyzes the profitability of the natural stone sector in Spain. Second, it studies the profitability of the natural stone sector in Italy. Third, it makes a reliable comparison between the two countries, identifying the common and differential aspects by analyzing the same time period with the same methodology and study variables.

The research is structured as follows. In section 2, a literature review of studies related to company profitability is conducted. Section 3 explains the sample selection and describes the variables used in the proposed model. Section 4 shows the results obtained, and section 5 discusses these results and presents the conclusions derived from this research.

## 2. Literature review and research hypotheses

Profitability is the result of efficient resource management and is the first objective of any organization. The study of profitability has been a topic of utmost importance for company stakeholders and researchers worldwide [35]. In particular, the factors explaining profitability as well as their significance have been of considerable relevance over the last decade. Many studies [9, 36–40] have analyzed the influence of several factors on the economic and financial profitability of companies. However, the results of these investigations differ depending on the country, industry type, and company size.

Previous literature has considered that profitability is determined by factors both internal and specific to the company, as well as by external factors, i.e., those elements of the environment that affect all companies within it. Subsequently, most research [41–44] has used information from company financial statements to explain variation in profitability. Other researchers [39, 45–49] have used, in addition to financial and accounting information, variables from the environment in which the company operates, such as geographical location and industry sector, as well as economic factors, such as inflation rate, country risk, and GDP.

Modigliani and Miller proposed, in 1958 [50], their capital structure irrelevance theory, which stated that the cost of capital and the value of a company are independent of its level of indebtedness. However, the theory was based on assumptions that were not in line with business reality, such as perfect capital markets with no transaction costs or taxes. In 1963, the tax effect was incorporated into the original thesis [51] such that, when the tax savings generated

by the cost of debt are taken into account, the financial structure is no longer neutral with respect to the value of the company and the cost of capital. More debt now implies a decrease in the weighted average cost of capital and a growth in firm value, such that more debt will always be preferred.

The incorporation of insolvency costs into Modigliani and Miller's 1963 thesis is called the Static Trade-Off Theory [21]. As the level of indebtedness increases, so does the probability of the company's insolvency. However, insolvency costs occur not only when the insolvency situation actually occurs, but also from the moment the company starts to become indebted, especially from the point at which the debt ratio exceeds a certain limit. This therefore means that the value of the company will increase with the level of indebtedness up to a certain limit, after which it will begin to decrease.

According to the Static Trade-Off Theory, the relationship between debt ratio and profitability will be positive, for two reasons: 1) high profits will allow for debt interest deductions, thereby putting upward pressure on indebtedness; 2) high profitability is interpreted as a sign of good company health, and hence implies a low probability of insolvency, thus promoting higher indebtedness. In practice, however, more profitable companies tend to have lower levels of debt. Consequently, the Pecking Order Theory appeared, which established that there is a hierarchy in the selection of sources of finance based on information asymmetries between managers and investors in the market [52, 53]. When a company needs resources, it will first make use of any available internal sources of finance. If this does not cover its needs, it will fall into debt and so will have to turn to the issuance of equity as a last resort. In short, whenever it is possible to use internal resources, there will be a negative relationship between profitability and indebtedness.

A company's assets can be tangible, intangible or financial, and represent its fundamental capacity to generate profitability, according to the Resource-Based View Theory. Those companies that possess more valuable and rare resources and capabilities will be able to achieve a competitive advantage and, therefore, higher financial performance in the short and long term. The effect will be greater if the competition cannot imitate these resources immediately [22, 54]. The growth of a company will be determined by both internal and external factors. Companies with higher growth rates will have more opportunities to increase their internal resources and, consequently, their future profitability [55].

The GDP variation rate results from a country's macroeconomic conditions and reflects the expansionary and recessionary cycles of the economy. In times of growth, consumers will have greater purchasing power, which will translate into greater sales and profits for companies. Conversely, in times of recession, sales will fall, with a consequent decline in business results [56].

With respect to inflation, another key macroeconomic variable, increased monetary instability is expected to reduce company profitability. Indeed, an increase in inflation will have a negative impact on costs and revenues. Rising prices will increase company costs. In turn, the purchasing power of customers will be reduced, with a consequent decrease in sales [3]. However, if a company has foreseen a rise in inflation, it could take measures to adjust its product prices and reduce operating costs, thereby increasing its results [57].

The board of directors is responsible for establishing a company's guiding strategies and is the main controlling body. Considering that the skills, abilities and characteristics of its members will influence its decisions, board composition is a topic of great interest in the literature, especially with regard to gender diversity [58, 59]. The relationship between the percentage of female board members and company profitability has been extensively studied in the literature, generally finding a positive and significant link between both variables [60]. The rationale is based on several theories, among which we can mention the following. According to Agency

Theory, female directors exert more stringent monitoring and decrease information asymmetries, thereby reducing agency costs and increasing profitability [23, 61]. The incorporation of female directors to a company's board can bring access to a larger number of resources according to the Resource Dependency Theory, thereby increasing company performance [24]. Based on the Stakeholder Theory, as women are more inclusive and have a greater propensity towards corporate social responsibility, gender-diverse boards are more likely to satisfy the demands of different stakeholders, thereby enhancing company reputation, stability in the market, and financial performance [25].

According to the literature, the degree of leverage is one of the most widely used indicators in profitability research. For Floros & Voulgaris (2016) and Almaqtari et al. (2019) [42, 62], indebtedness does not affect profitability. In contrast, Rahman et al. (2020), Alarussi & Alhaderi (2018), and Asimakopoulos et al. (2009) [9, 63, 64] conclude that the degree of leverage has a negative effect on profitability. On the contrary, Singapurwoko and El-Wahid (2011) [65] and Becker-Blease et al. (2010) [66] state that debt has a positive impact on ROA but a negative effect on ROE.

Company size has been shown to be positively related to profitability in studies such as those conducted by Asimakopoulos et al. (2009) and Asche et al. (2018) [28, 64]. Kouser et al. (2012), Koralun-Bereźnicka & Ciołek (2018), and Becker-Blease et al. (2010) [8, 16, 67] conclude that the influence is negative. However, other works argue that there is no relationship between the two variables [29, 68].

In addition to these indicators, some researchers explored the relationship of other variables such as stock turnover [69–73], asset turnover [14, 18, 74–76], company growth [2, 13, 16, 77–79], company age [3, 37, 78, 80–82], inflation rate, and GDP [3, 38, 39, 62, 83, 84]. However, the extent of the impact of the various factors and their relationship with profitability do not coincide. A general limitation in most of these studies is that they refer to all companies in general and not to those within a specific sector. However, sectors can show important differences between one another.

Based on the above, the following hypotheses were stated in this research:

Hypothesis 1 (H1): *Company size has a significant positive effect on profitability.*

Hypothesis 2 (H2): *The degree of indebtedness has a significant negative impact on profitability.*

Hypothesis 3 (H3): *Company growth has a positive and significant impact on profitability.*

Hypothesis 4 (H4): *GDP variation positively and significantly impacts profitability.*

Hypothesis 5 (H5): *Inflation has a negative and significant effect on profitability generation.*

Hypothesis 6 (H6): *The percentage of women in managerial positions is positively and significantly related to company profitability.*

## 3. Methodology

### 3.1. Sample selection and data collection

Accounting information from the financial statements of all companies grouped according to NACE code 0811 (National Classification of Economic Activities: extraction of ornamental and building stone, limestone, gypsum, chalk, and slate) in Spain and Italy was used. For this purpose, a time horizon was set covering five consecutive years from 2015 to 2019. The year 2015 was chosen as the first year of the study because we did not want the conclusions to be outdated, considering the cycles of the economy. The study ended in 2019, as 2020 was a highly atypical year for companies, especially in Spain and Italy, two of the countries most

heavily hit by the COVID-19 pandemic, with many months of paralyzed economic activity. Therefore, including the year 2020 in the analysis would have distorted the results. The period 2015–2019 represents a 5-year period of normal economic activity, thereby allowing reliable conclusions to be drawn.

The final sample contains 453 companies, of which 203 are Spanish and 250 Italian. Finally, 2,136 valid observations were obtained, 941 for Spain and 1,195 for Italy. The companies' financial data were extracted from the Amadeus database of Bureau Van Dijk [85]. The study sample comprises all those companies included in Amadeus under the NACE code 0811, for which the database provides data on the variables used.

## 3.2. Description of variables

This research has focused on examining the factors that, to a greater extent, determine the profitability of companies operating in the natural stone sector. For the study of company performance, the most commonly used variables are ROA (*return on assets*—economic profitability) and ROE (*return on equity*—financial profitability), due to their ability to measure investments in terms of assets and equity [9, 10, 86, 87]. Previous research [2, 4, 62, 88] has used ROA and ROE as dependent variables. Therefore, in the two empirical analyses implemented in this study, ROA and ROE were considered as dependent variables. Both are continuous quantitative variables.

The explanatory variables under analysis can be grouped into three distinct categories: (i) those associated with the company i.e., the age of the company in the market, as well as financial variables such as the volume of assets, leverage, total operating income, stock turnover, asset turnover, average collection period, average payment period, company growth, and legal form; (ii) those associated with the economic environment i.e., country, gross domestic product (GDP), and level of inflation; (iii) those linked to diversity in business management, identified in this research by the gender variable.

The following independent variables were considered in the study: company size, measured by the volume of assets; the degree of leverage; company growth; the change in the country's GDP; inflation; and the percentage of female board directors. Finally, the following were used as control variables: operating income, stock turnover, asset turnover, average recovery and payment periods, company age, and legal form.

Table 1 shows the description of all variables used in the empirical analysis.

**3.2.1. Financial profitability.** The return on equity (ROE) indicator expresses a company's ability to generate profits through a productive use of shareholders' contributions and efficient management. It is calculated as the ratio of the company's net profit after tax to shareholders' equity. This indicator has been widely used in studies such as Al-Jafari & Alchami (2014), Alarussi & Alhaderi (2018), Banerjee (2015), Burja (2011), and Rahman et al. (2020) [9, 36, 38, 63, 89].

**3.2.2. Economic profitability.** Return on assets (ROA) is defined as the company's net profit after tax divided by total assets, and has also been widely used in the literature [2, 10, 65, 81, 90].

**3.2.3. Company size.** Company size is often considered an important factor when explaining profitability [67, 91]. The theory suggests that larger companies are more likely to access financial markets and obtain better interest rates by exploiting economies of scale. Currently, researchers disagree on the definition of company size. Some studies define it in terms of total assets, total operating income or number of employees [68, 78, 91].

For Y. S. Chen & Chang (2010), Budisaptorini et al. (2019), and Akinlo (2012) [26–27, 92], company size has a positive and significant effect on profitability. In contrast, authors such as

**Table 1. Definition of variables.**

| Abbreviation | Variable | Definition |
|---|---|---|
| ROE | Return on Equity | Net profit divided by equity |
| ROA | Return on Assets | Net profit divided by total assets |
| Size | Company size | Natural logarithm of total assets in the company |
| Debt | Indebtedness | Total liabilities divided by total assets |
| Growth | Company growth | Percentage change in total assets |
| VarGDP | Change in GDP | Percentage change in gross domestic product |
| Inflat | Inflation | The country's inflation for the year |
| Gender | Gender diversity | Percentage of female board directors |
| OpInc | Operating income | Natural logarithm of operating income |
| StockT | Stock turnover | Cost of sales divided by stock |
| StockT | Asset turnover | Operating income divided by total assets |
| ARP | Average recovery period | The average number of days the company takes to receive payment from customers |
| APP | Average payment period | The average number of days the company takes to pay suppliers |
| Age | Company age | Age of the company in years |
| LForm | Legal form | The legal form of the company:<br>• Public limited company<br>• Private limited company<br>• Cooperative<br>• Other legal forms |
| Country | Company's country of residence | Dummy variable, equal to 1 for Spain and 0 for Italy |

Source: Own elaboration.

Asche et al. (2018), Evans (1987), and Becker-Blease et al. (2010) [28, 67, 78] claim that company size, defined as the level of assets, has a negative relationship with profitability. Other research finds that size has no impact on company performance [29, 68]. According to this research, size is defined as the contingent of assets controlled by the company, and its relationship with profitability is expected to be positive, as stated in Hypothesis 1.

**3.2.4. Indebtedness.**   Indebtedness is one of the most critical factors in analyzing both corporate performance and its impact on company performance. The debt ratio is defined as a company's total debts divided by its total assets. Previous research results generally find an inverse relationship between the level of indebtedness of a company and its profitability [4, 44, 48, 90, 93, 94], as reflected in Hypothesis 2.

**3.2.5. Growth.**   Some previous research used company growth as the percentage change in operating income [2, 13, 16, 35]. However, as in this study, other researchers consider company growth to be the percentage change in total assets [29, 55, 95, 96]. These research results show a positive and significant relationship between the percentage change in assets and company profitability, consistent with Hypothesis 3.

**3.2.6. GDP variation.**   GDP is one of the most widely used indicators for measuring economic activity within a country. Economic growth reflects general macroeconomic conditions. It is presumed that a change in GDP can influence company performance. Demand for goods and services increases during economic growth cycles, and so companies are expected to increase sales and thus profitability. Conversely, during periods of economic recession, company performance deteriorates. This macroeconomic variable has been analyzed in several studies [39, 45–48, 97–100]. Results from previous work [3, 45, 46, 49] show that economic growth has a positive and significant impact on firm performance. However, empirical

evidence also shows, in some cases, that the relationship between profitability and economic growth is negative [47, 100]. The research has also concluded that there is no statistically significant relationship between economic growth and profitability [39, 97]. According to most of the literature, as stated in Hypothesis 4, a positive relationship between GDP variation and profitability is expected.

**3.2.7. Inflation.** The inflation rate is another commonly used macroeconomic indicator in profitability studies. It is defined as the rate at which the general price level of goods and services increases, leading to a decrease in purchasing power [62].

The effect of inflation on a company's profitability will depend on whether inflation is anticipated or unanticipated [57]. In the case of anticipated inflation, companies can ensure that costs do not exceed revenues by adjusting the prices of goods and services beforehand. Therefore, some researchers [3, 39, 46, 47, 100] conclude that inflation positively and significantly affects firm profitability. Conversely, when inflation is unanticipated, companies are not able to make appropriate price adjustments, leading to an increase in costs compared to revenues and, hence, a decrease in profitability. This is the reasoning behind Hypothesis 5, which states that there is a negative relationship between inflation and profitability.

**3.2.8. Gender.** Many studies in the literature have analyzed the influence of gender-diverse boards of directors on company profitability. Most have concluded that a greater female presence has a positive influence on profitability [60, 101–104]. However, some research has found an inverse relationship [105, 106] or even no relationship at all [59, 107, 108].

**3.2.9. Operating income.** A company's operating income is considered to be a key indicator of many positive aspects that support both growth and profitability. Previous research shows a positive relationship between operating income and firm performance [13, 28, 64, 77]. This study uses the natural logarithm of operating income to determine its relationship with firm profitability.

**3.2.10. Stock turnover.** This ratio is an important measure for assessing management efficiency in converting inventory into sales. A high stock turnover is generally indicative of efficient inventory management. On the other hand, overstocking in the product line can cause inventory turnover to decrease. Authors such as Nageswararao et al. (2019), Thi et al. (2020), Gołaś (2020), and Otekunrin et al. (2021) [69, 109, 110] consider inventory turnover to be a measure of working capital and to have a positive relationship with profitability.

**3.2.11. Asset turnover.** Asset turnover, considered a fundamental indicator of corporate governance, is a financial ratio that measures the efficiency of a firm's asset use in generating operating income. The results of previous work disagree on the relationship between asset turnover and company profitability [111]. For Shahnia et al. (2020) [4], asset turnover has no significant impact on return on assets. On the other hand, Abdulla (2020) and Akoto et al. (2013) [14, 66] point out that the relationship of this indicator with profitability is positive.

**3.2.12. Average recovery period.** The average collection period represents the average number of days the company takes to collect payment after making a credit sale. Previous research [72] indicates that there exists a positive and significant relationship between the average collection period and profitability. In contrast, other authors [71, 112–114] have found that this indicator has a negative impact on profitability.

**3.2.13. Average payment period.** This indicator denotes the average number of days a firm takes to pay its current or short-term debts. Empirical studies have found a negative relationship between the average payment period and company profitability [15, 71, 72, 112, 114]. On the other hand, Raza et al. (2015), Kumaraswamy (2016) and Ngwenya (2010) [113, 115, 116] conclude that this indicator has a positive relationship with company performance.

**3.2.14. Age.** In previous research based on the life cycle of the company, the relationship between the age of a company and its performance in terms of profitability is complex [15, 37,

74, 117]. The age of a company is quantified by the number of years it has been in the market [82]. Many studies [3, 79, 80] conclude that there is a significant and positive relationship between company age and profitability. In contrast, Adekunle (2011), Brooks & Buckmaster (1976), Fairfield et al. (1996), and Freeman et al. (1982) [43, 118–120] argue that company age has a negative effect on profitability.

**3.2.15. Legal form.** Creixans-Tenas & Arimany-Serrat (2018) [121] dichotomize legal form, considering limited and limited liability companies in order to explain profitability. However, their study on Spanish private hospitals found no significant relationship between legal form and profitability.

**3.2.16. Country.** The country where the company is located can significantly influence profitability [122], as the industry can have differentiating characteristics depending on its geographical location [58, 123].

### 3.3. Methodology

First, a linear correlation analysis was applied to determine which variables were significantly correlated with profitability and to discard high correlations between the regressors. A multiple linear regression analysis was then implemented to identify the relationship between the explanatory variables and profitability [9, 10, 54, 124–126]. In addition, to address any possible endogeneity in the proposed model, and in accordance with the literature, the dependent variable, with a lag of one and two periods [54, 83, 127], was used as a regressor [128, 129].

In addition, the panel data methodology, which combines time-series and cross-sectional data, was used to eliminate possible unobservable heterogeneity across the firms in the sample and to control for omitted variables in the empirical study. The fixed-effects estimation model is more appropriate when there is unobservable heterogeneity across firms correlated with the regressors. Otherwise, a random-effects estimation model is the preferred method. The Hausman test was used [130] to determine which model provides the most consistent estimators.

The goodness of fit of each model was assessed using the $F$ statistic, which analyses the joint significance of the regressors, and the adjusted $R^2$, which shows the proportion of the variation in the dependent variable that is explained by the set of regressors. A comparison was made between the different models using the Akaike information criterion (AIC) and the Bayesian information criterion (BIC), with smaller values indicating the best models [131, 132].

## 4. Results

### 4.1. Descriptive statistics and correlations

Table 2 shows the main descriptive statistics of the variables used to explain the profitability of companies operating in the natural stone sector in Spain and Italy from 2015 to 2019. The data are presented for both the total sample and each subsample according to the country where the company is located. It also includes the test of means, which shows significant differences between the two countries.

Financial profitability in this sector is lower in Spain, with an average of 5.88%, compared to Italy, with 8.74%, Malaysia, with 6.74% [9], and India, with 10.16% [62].

The average economic profitability of Spanish companies is 2.96%, while the average return of Italian companies is 3.19%. Both values are higher than those generally shown by Indian companies (0.72% in Gaur & Mohapatra, 2021; 1.17% in Almaqtari et al., 2019) [133], and by Polish companies (0.92% in Anton & Afloarei Nucu, 2020) [124]. However, this indicator is below that of Indonesia (3.57% in Shahnia et al., 2020) [4]. In sum, the differences in ROA between Spain and Italy are small and non-significant.

**Table 2. Descriptive statistics and mean difference test by country.**

| Variable | Total sample | | | | Spain | | | | Italy | | | | p-value+ |
|---|---|---|---|---|---|---|---|---|---|---|---|---|---|
| | Mean | Standard deviation | Minimum | Maximum | Mean | Standard deviation | Minimum | Maximum | Mean | Standard deviation | Minimum | Maximum | |
| ROE | 7.48 | 29.23 | -274.84 | 270.22 | 5.88 | 29.39 | -267.38 | 176.74 | 8.74 | 59.05 | -274.84 | 270.22 | 0.0243** |
| ROA | 3.09 | 7.43 | -44.59 | 41.86 | 2.96 | 7.56 | -43.85 | 41.86 | 3.19 | 7.33 | -44.59 | 41.10 | 0.4652 |
| Size | 8.26 | 1.08 | 2.75 | 12.63 | 8.26 | 1.22 | 3.95 | 12.63 | 8.25 | 0.95 | 2.75 | 12.26 | 0.8235 |
| Debt | 0.53 | 0.30 | 0.00 | 2.59 | 0.44 | 0.29 | 0.01 | 1.86 | 0.61 | 0.29 | 0.00 | 2.59 | 0.0000*** |
| Growth | 0.41 | 11.09 | -0.53 | 454.58 | 0.10 | 0.48 | -0.48 | 7.1 | 0.66 | 14.81 | -0.53 | 454.58 | 0.2985 |
| VarGDP | 1.81 | 1.07 | 0.29 | 3.84 | 2.84 | 0.64 | 0.29 | 3.84 | 0.99 | 0.47 | 0.29 | 1.67 | 0.0000*** |
| Inflat | 0.66 | 0.77 | -0.50 | 1.96 | 0.74 | 0.98 | -0.50 | 1.96 | 0.60 | 0.54 | -0.09 | 1.23 | 0.0000*** |
| Gender | 21.52 | 30.88 | 0.00 | 100 | 24.41 | 30.01 | 0.00 | 100 | 19.25 | 31.37 | 0.00 | 100 | 0.0001*** |
| OpInc | 7.43 | 1.42 | 0.00 | 12.43 | 7.51 | 1.36 | 0.00 | 12.43 | 7.37 | 1.46 | 0.00 | 11.92 | 0.0211** |
| StockT | 67.69 | 328.96 | 0.00 | 9680.32 | 50.11 | 237.86 | 0.00 | 3647.51 | 81.51 | 385.38 | 0.00 | 9680.32 | 0.0285** |
| AssetT | 0.68 | 0.68 | 0.00 | 15.20 | 0.75 | 0.87 | 0.00 | 15.20 | 0.63 | 0.46 | 0.00 | 3.69 | 0.0001*** |
| ARP | 141.57 | 145.26 | 0.00 | 981.75 | 143.41 | 133.06 | 0.00 | 967.13 | 140.13 | 154.23 | 0.00 | 981.75 | 0.6043 |
| APP | 67.67 | 105.11 | 0.00 | 993.43 | 50.72 | 79.69 | 0.00 | 993.43 | 81.02 | 119.80 | 0.00 | 962.24 | 0.0000*** |
| Age | 28.50 | 16.91 | 0.14 | 103.41 | 26.96 | 12.31 | 0.84 | 62.05 | 29.72 | 19.72 | 0.14 | 103.41 | 0.0002 |

Number of observations: 2136 in the total sample, 941 in Spain and 1195 in Italy. In Growth, as there is a lag due to the difference between years, the number of observations is 1,715, 941 and 962, respectively.

+ Mean difference test.

***, ** and * denote a significance level below 1%, 5% and 10%, respectively.

Source: Own elaboration.

Company size, measured by the volume of assets, is similar for both countries. However, company growth is higher in Italy, although the difference is not significant. The same is true for age, with an average of 28.50 years for the companies in the sector.

Indebtedness for the sector is significantly higher in Italy, with 61% borrowed funds as a proportion of total financing, while Spain has an average debt of 44%, relying less on external funding. In contrast, operating revenues are significantly higher in Spain.

Indicators considered part of working capital [70, 115, 134] were also analyzed, such as stock turnover, which is significantly higher in Italy. However, the reverse is true for asset turnover, which is significantly higher in Spain. Regarding the payment period, Spanish firms take over 50 days to service their debts, while Italian firms take more than 81 days. However, there is no significant difference in the average collection period between Spanish and Italian companies.

Concerning the gender variable, Spain has, on average, significantly more women in top management roles (24.41%) within the sector than Italy (19.25%), which indicates that Spanish companies in the natural stone industry are more open to incorporating female board members than Italian companies. However, if we compare the ratio of men and women on the boards of directors, a marked difference is evident, which may be attributed to the nature of the activities carried out by these companies. Some researchers [135–137] consider this disproportionate distribution of women and men by specific job sectors as horizontal segregation.

As far as macroeconomic variables are concerned, there are significant differences between the two countries, despite their geographical proximity within the European Union. In the period analyzed, the growth of the Spanish economy has been visibly higher than in Italy. While Spain's GDP grew by an average of 2.84%, Italy's grew by only 0.99%. Moreover, inflation in Spain was 0.14% higher than in Italy.

**Table 3. Legal form by country.**

| Legal form | Total sample | Spain | Italy | Chi-squared test [+] |
|---|---|---|---|---|
| Public limited company | 379 | 300 | 79 | 251.8833 (0.0000) |
| Private limited company | 1664 | 625 | 1039 | |
| Cooperative | 58 | 16 | 42 | |
| Other legal form | 35 | 0 | 35 | |

[+] *p*-value in brackets.

Source: Own elaboration.

Table 3 shows the results of the chi-squared test applied to the legal form according to the country of reference. Significant differences are found between Spain and Italy. While there are many companies in Spain with the legal form of a public limited company, there is a clear preference for private limited companies in Italy. Similarly, Italy has cooperatives and other social forms in this sector which hardly exist at all in Spain. Indeed, Spain is a country where the cooperative legal form is widely used in other sectors, such as agriculture, but is almost absent within the natural stone sector.

Table 4 shows the Pearson correlation matrix between the continuous variables used in the empirical study. It can be seen that there are no high correlations between the regressors that could give rise to collinearity problems in the subsequent multivariate analysis.

Moreover, all regressors except inflation show a significant correlation with the dependent variables (ROE and ROA). Specifically, volume of assets, average collection and payment periods, company age, change in GDP, and management gender are negatively correlated with ROE and ROA. For change in GDP and gender, the relationship is only significant for ROE. In contrast, the correlation is positive for the variables measuring operating income, stock turnover, asset turnover, and growth, although only the correlation between ROA and growth is significant. Concerning indebtedness, the correlation is significant and negative for ROA but positive for ROE.

Table 5 shows the analysis results of variance of ROA and ROE as a function of legal form. It can be seen that there is indeed a significant relationship between the company's legal form and the two types of return, in both Spain and Italy.

## 4.2. Multivariate analysis

Table 6 shows the panel data and ordinary least squares regression analysis for financial profitability. Both in the total sample and the subsamples for Spain and Italy, the fixed-effects model (reported) was better than the random-effects model (not reported), as the Hausman test yielded a *p-value* of less than 0.05 in all cases. Furthermore, the fixed effects model outperformed the ordinary least squares model, as stated by the Breusch-Pagan test (*p-value* $< 0.05$), and the AIC and BIC criteria values.

The results show that larger companies with higher growth (measured as asset growth), higher asset turnover, and lower debt levels achieve higher profitability. Moreover, in the case of Italy, the change in GDP has a positive and significant effect on profitability. The other variables were not significant. The size of the coefficients indicates that the most influential variable is the level of indebtedness. With the proposed model, we explained 40.38% of the ROE of Spanish companies and 55.02% of Italian companies.

Table 7 shows the results for the economic profitability analysis. As before, the fixed-effects model outperforms the random-effects and ordinary least squares models. We again find that company size, company growth, and asset turnover positively and significantly influence

**Table 4. Pearson correlations between the continuous variables.**

| | ROE | ROA | Size | Debt | Growth | VarGDP | Inflat | Gender | OpInc | StockT | AssetT | ARP | APP |
|---|---|---|---|---|---|---|---|---|---|---|---|---|---|
| **ROA** | 0.5523*** (0.0000) | | | | | | | | | | | | |
| **Size** | -0.1203*** (0.0000) | -0.0661*** (0.0022) | | | | | | | | | | | |
| **Debt** | 0.0502* (0.0203) | -0.1994*** (0.0000) | -0.2082*** (0.0000) | | | | | | | | | | |
| **Growth** | 0.0301 (0.2127) | 0.0409* (0.0901) | 0.0040 (0.8692) | -0.0063 (0.7949) | | | | | | | | | |
| **VarGDP** | -0.0457** (0.0345) | -0.0296 (0.1718) | -0.0114 (0.6000) | -0.2270*** (0.0000) | -0.0124 (06068) | | | | | | | | |
| **Inflat** | -0.0010 (0.9624) | 0.0289 (0.1821) | 0.0292 (0.1766) | -0.0330 (0.1271) | -0.0313 (0.1950) | -0.0478** (0.0270) | | | | | | | |
| **Gender** | -0.0388* (0.0732) | -0.0208 (0.3358) | -0.0542** (0.0123) | -0.0416* (0.0545) | -0.0192 (0.4268) | 0.0720*** (0.0009) | 0.0028 (0.8988) | | | | | | |
| **OpInc** | 0.01166*** (0.0000) | 0.2327*** (0.0000) | 0.5535*** (0.0000) | -0.1130*** (0.0000) | 0.0001 (0.9977) | 0.0176 (0.4154) | 0.0398* (0.0656) | -0.0618*** (0.0043) | | | | | |
| **StockT** | 0.0944*** (0.0000) | 0.1194*** (0.0000) | -0.0107 (0.6225) | 0.0175 (0.4191) | -0.0022 (0.9273) | -0.0523** (0.0156) | -0.0301 (0.1637) | -0.0067 (0.7579) | 0.0575*** (0.0079) | | | | |
| **AssetT** | 0.2657*** (0.0000) | 0.2984*** (0.0000) | -0.4222*** (0.0000) | 0.1674*** (0.0000) | -0.0002 (0.9936) | 0.0673*** (0.0018) | 0.0140 (0.5177) | -0.0210 (0.3314) | 0.2343*** (0.0000) | 0.0812*** (0.0002) | | | |
| **ARP** | -0.0889*** (0.0000) | -0.1818*** (0.0000) | 0.0844*** (0.0001) | -0.0218 (0.3140) | 0.0028 (0.9092) | 0.0145 (0.5033) | -0.0226 (0.2958) | -0.0415* (0.0553) | -0.1334*** (0.0000) | -0.0099 (0.6477) | -0.2505*** (0.0000) | | |
| **APP** | -0.0713*** (0.0010) | -0.1794*** (0.0000) | -0.0330 (0.1271) | 0.1787*** (0.0000) | 0.0037 (0.8772) | -0.1243*** (0.0000) | -0.0360* (0.0966) | -0.0570*** (0.0084) | -0.1610*** (0.0000) | -0.0305 (0.1594) | -0.1532*** (0.0000) | 0.2823*** (0.0000) | |
| **Age** | -0.1115*** (0.0000) | -0.0659*** (0.0023) | 0.3205*** (0.0000) | -0.1683*** (0.0000) | -0.0542** (0.0248) | -0.0941*** (0.0000) | 0.0244 (0.2587) | -0.0599*** (0.0056) | 0.1540*** (0.0000) | -0.0413* (0.0565) | -0.2061*** (0.0000) | 0.0000 (0.9983) | -0.0714*** (0.0010) |

$p$- value in brackets.

***, **, and * denote a significance level below 1%, 5% and 10%, respectively.

Number of observations: 2136, except for Growth, with 1,715.

Source: Own elaboration.

**Table 5. Average ROA and ROE by legal form.** Analysis of variance.

| Legal form | Total sample | | Spain | | Italy | |
|---|---|---|---|---|---|---|
| | ROE | ROA | ROE | ROA | ROE | ROA |
| Public limited company | 3.06 | 1.94 | 2.67 | 1.80 | 4.56 | 2.50 |
| Private limited company | 8.76 | 3.44 | 7.49 | 3.55 | 9.52 | 3.37 |
| Cooperative | 4.24 | 1.86 | 2.91 | 1.39 | 4.74 | 2.04 |
| Other legal form | -0.02 | 0.73 | | | -0.02 | 0.73 |
| F | 4.98*** (0.0019) | 5.97*** (0.0005) | 2.83* (0.0595) | 5.86*** (0.0030) | 2.13* (0.0952) | 2.11* (0.0975) |

*p*-value in brackets.

***, ** and * denote a significance level below 1%, 5% and 10%, respectively.

Source: Own elaboration.

profitability. In contrast, firm leverage has a negative and significant influence on profitability in both Spain and Italy. Moreover, in Italy, the average payment period has a significantly negative relationship with profitability. In the total sample, company age has a negative and significant relationship with profitability, although this was not confirmed in the country subsamples. However, considering the size of the coefficient, the effect is small. The model explained 59.30% of ROA in Spanish companies and 73.52% in Italian companies.

Therefore, in both countries, the results obtained confirmed Hypotheses 1 (positive asset ratio), 2 (negative debt ratio), and 3 (positive growth ratio).

Regarding Hypothesis 4, it was confirmed that change in GDP has a direct relationship with profitability, but only for the case of financial profitability in Italian companies.

Inflation showed a negative sign in the coefficients of both models for Spain and Italy, in the case of ROE and ROA. However, no significance was found, so Hypothesis 5 was not confirmed.

Finally, with respect to gender, the coefficient was negative for Spain and positive for Italy for both ROE and ROA, and was non-significant, so Hypothesis 6 was not confirmed for this sector in the period analyzed.

## 5. Discussion and conclusions

The descriptive analysis showed that the average financial return of Spanish companies is 2.86% lower than that of Italian companies. However, compared to companies from countries such as Malaysia [9] and India [62], Italian companies have a lower return on equity.

Moreover, Italian companies have a slightly higher return on assets (0.23%) than Spanish companies. Other studies show that, in countries such as India [62, 133], companies have lower economic profitability than in Italy and Spain.

One striking aspect is the percentage of women in the top management positions. This study shows that the percentage of female board members of Spanish companies within the natural stone sector is 24.41%, compared to 19.25% for Italian companies. However, according to the Global Gender Gap Report 2020, the percentage of female board members across all sectors of activity in Spain was 22%, compared to 34% for Italy. In other words, the natural stone sector presents an inverse ranking between the two countries with respect to the rest of the sectors. It is worth noting that, in Italy, the ratio of women in the industry is 14.75% lower than in the business sector as a whole. Perhaps this disproportionate distribution is due to the type of

**Table 6. ROE regression analysis.**

| Variable | Total sample | | Spain | | Italy | |
|---|---|---|---|---|---|---|
| | OLS | FE | OLS | FE | OLS | FE |
| **Intercept** | -9.6568 (0.333) | 198.4072 (0.552) | -24.2771 (0.202) | 318.7547 (0.863) | -4.5859 (0.747) | -125.0431** (0.040) |
| **ROEret1** | 0.2210*** (0.000) | -0.3123*** (0.000) | 0.1636*** (0.000) | -0.2806*** (0.000) | 0.2993*** (0.000) | -0.3661*** (0.000) |
| **ROEret2** | 0.1046*** (0.000) | -0.1625*** (0.000) | 0.1187** (0.010) | -0.0923* (0.086) | 0.0804** (0.018) | -0.2140*** (0.000) |
| **Size** | 0.7336 (0.574) | 18.6234*** (0.002) | -0.3249 (0.872) | 19.5240** (0.040) | 0.5384 (0.772) | 18.51879** (0.015) |
| **Debt** | -7.0020*** (0.006) | -99.2230*** (0.000) | -9.4611** (0.025) | -96.4061*** (0.000) | -6.2417* (0.070) | -103.5897*** (0.000) |
| **Growth** | 15.5753*** (0.000) | 15.0523*** (0.000) | 11.3406*** (0.001) | 11.5748*** (0.004) | 26.3350*** (0.000) | 19.9867*** (0.000) |
| **VarGDP** | 2.0389 (0.383) | -7.4033 (0.581) | 7.5160 (0.232) | -15.2686 (0.876) | 0.4187 (0.911) | 5.8722** (0.047) |
| **Inflat** | -3.4785 (0.237) | -7.8017 (0.139) | -6.8864 (0.178) | -7.9880 (0.473) | -1.2309 (0.874) | -4.7898 (0.427) |
| **Gender** | 0.0005 (0.980) | | -0.0337 (0.351) | | 0.0279 (0.320) | |
| **OpInc** | 0.3470 (0.694) | 3.0878 (0.137) | 1.6583 (0.278) | -0.7069 (0.841) | -0.3661 (0.756) | 3.5596 (0.172) |
| **StockT** | 0.0034 (0.240) | 0.0009 (0.814) | 0.0016 (0.723) | -0.0019 (0.763) | 0.0051 (0.171) | 0.0015 (0.739) |
| **AssetT** | 9.8624*** (0.000) | 18.5829*** (0.000) | 10.1142*** (0.002) | 14.6888** (0.038) | 10.1091*** (0.007) | 30.7109*** (0.000) |
| **ARP** | -0.0013 (0.807) | -0.0088 (0.367) | 0.0033 (0.718) | 0.0067 (0.708) | -0.0021 (0.733) | -0.0175 (0.128) |
| **APP** | -0.0132* (0.073) | -0.0138 (0.303) | -0.0080 (0.600) | -0.0028 (0.929) | -0.0127 (0.128) | -0.0118 (0.404) |
| **Age** | -0.0417 (0.347) | -10.2003 (0.320) | 0.1711 (0.138) | -13.7869 (0.806) | -0.0500 (0.295) | -0.5272 (0.902) |
| **Legal form Cooperative** | -2.8447 (0.678) | | | | -5.2077 (0.466) | |
| **Private** | 6.7478 (0.203) | | 3.9704 (0.663) | | 5.7756 (0.271) | |
| **Public** | 4.6663 (0.409) | | -1.3861 (0.878) | | 6.4356 (0.303) | |
| **Country** | -4.1766 (0.156) | | | | | |
| **Observations** | 1216 | 1216 | 529 | 529 | 687 | 687 |
| **Adjust $R^2$** | 0.1939 | 0.4826 | 0.1563 | 0.4038 | 0.2402 | 0.5502 |
| **F** | 17.24*** (0.0000) | 18.19*** (0.0000) | 7.11*** (0.0000) | 6.13*** (0.0000) | 13.76*** (0.0000) | 15.54*** (0.0000) |
| **Breusch-Pagan** | | 2.567 (0.000) | | 2.132 (0.000) | | 2.963 (0.000) |
| **Hausman test** | | 490.81 (0.0000) | | 50.83 (0.0000) | | 288.77 (0.0000) |
| **AIC** | 11131.89 | 10046.29 | 4890.63 | 4458.91 | 6240.54 | 5573.17 |
| **BIC** | 11228.85 | 10117.74 | 4963.23 | 4518.71 | 6322.12 | 5632.09 |

*p*-value in brackets.

\*\*\*, \*\* and \* denote a significance level below 1%, 5% and 10%, respectively.

AIC and BIC smaller is better.

Source: Own elaboration.

**Table 7. ROA regression analysis.**

| Variable | Total sample | | Spain | | Italy | |
|---|---|---|---|---|---|---|
| | OLS | FE | OLS | FE | OLS | FE |
| **Intercept** | -2.4997 (0.292) | -23.1582 (0.757) | -5.1221 (0.277) | -64.6408 (0.878) | -1.4719 (0.645) | -51.9183*** (0.000) |
| **ROAret1** | 0.3729*** (0.000) | -0.2825*** (0.000) | 0.2628*** (0.000) | -0.3044*** (0.000) | 0.4707*** (0.000) | -0.2418*** (0.000) |
| **ROAret2** | 0.1804*** (0.000) | -0.2043*** (0.000) | 0.1504*** (0.000) | -0.2507*** (0.000) | 0.2062*** (0.000) | -0.1553*** (0.001) |
| **Size** | 0.1745 (0.574) | 7.3229*** (0.000) | -0.3838 (0.444) | 7.1077*** (0.001) | 0.2187 (0.601) | 7.4238*** (0.000) |
| **Debt** | -2.8268*** (0.000) | -25.7397*** (0.000) | -4.6362*** (0.000) | -32.1784*** (0.000) | -2.1498*** (0.009) | -21.2167*** (0.000) |
| **Growth** | 4.2405*** (0.000) | 3.0538*** (0.000) | 3.3507*** (0.000) | 2.2807** (0.013) | 6.9702*** (0.000) | 4.4846*** (0.000) |
| **VarGDP** | 0.7242 (0.192) | -0.1013 (0.973) | 1.3026 (0.403) | 1.9829 (0.929) | 0.4111 (0.627) | 0.7931 (0.216) |
| **Inflat** | -0.7126 (0.309) | -0.9199 (0.437) | -1.1701 (0.356) | -0.8286 (0.745) | -0.1387 (0.937) | -0.1889 (0.886) |
| **Gender** | 0.0010 (0.856) | | -0.0015 (0.864) | | 0.0026 (0.680) | |
| **OpInc** | 0.1871 (0.373) | 0.6068 (0.188) | 0.9369** (0.014) | 1.0173 (0.206) | -0.1619 (0.542) | 0.1566 (0.778) |
| **StockT** | 0.0012* (0.088) | 0.0001 (0.951) | 0.0008 (0.492) | -0.0004 (0.777) | 0.0016* (0.054) | 0.0004 (0.665) |
| **AssetT** | 2.2897*** (0.000) | 7.4481*** (0.000) | 2.0823*** (0.008) | 5.9136*** (0.000) | 2.2215*** (0.008) | 9.5983*** (0.000) |
| **ARP** | -0.0004 (0.758) | -0.0019 (0.391) | 0.0012 (0.595) | 0.0015 (0.716) | -0.0002 (0.856) | -0.0024 (0.325) |
| **APP** | -0.0132* (0.073) | -0.0046 (0.120) | -0.0011 (0.775) | 0.0074 (0.309) | -0.0047** (0.013) | -0.0073** (0.018) |
| **Age** | -0.0044** (0.013) | -0.9226 (0.689) | 0.0085 (0.764) | 0.2759 (0.983) | 0.0086 (0.419) | 0.0326 (0.973) |
| **Legal form Cooperative** | -1.3879 (0.392) | | | | -1.8068 (0.258) | |
| **Private** | 1.5215 (0.226) | | 2.2706 (0.314) | | 1.1384 (0.333) | |
| **Public** | 0.8448 (0.528) | | 0.7701 (0.731) | | 1.5222 (0.279) | |
| **Country** | -1.1908* (0.089) | | | | | |
| **Observations** | 1216 | 1216 | 529 | 529 | 687 | 687 |
| **Adjust $R^2$** | 0.4214 | 0.6721 | 0.3260 | 0.5930 | 0.5215 | 0.7352 |
| **F** | 50.15*** (0.0000) | 22.92*** (0.0000) | 16.96*** (0.0000) | 10.49*** (0.0000) | 44.98*** (0.0000) | 15.00*** (0.0000) |
| **Breusch-Pagan** | | 3.162 (0.000) | | 2.788 (0.000) | | 3.299 (0.000) |
| **Hausman test** | | 649.84 (0.0000) | | 258.83 (0.0000) | | 342.23 (0.0000) |
| **AIC** | 7638.95 | 6401.68 | 3412.75 | 2897.87 | 4192.65 | 3478.82 |
| **BIC** | 7735.91 | 6473.13 | 3485.36 | 2957.66 | 4274.23 | 3537.74 |

$p$-value in brackets.

***, ** and * denote a significance level below 1%, 5% and 10%, respectively.

AIC and BIC smaller is better.

Source: Own elaboration.

activity carried out by companies in this sector [137]. In any case, unlike other industries [60], the impact of management gender on profitability is not significant in the natural stone sector.

On average, company size is slightly larger in Spain than in Italy, as are operating income and asset turnover. However, recourse to external financing is significantly higher in Italy. The financial indicators of stock turnover and average payment period, considered part of working capital [138–140], are significantly higher in Italy than in Spain. However, the average collection period is similar for both countries.

With respect to macroeconomic variables, the Spanish economy shows better growth than the Italian economy, despite the latter having a lower inflation rate.

This research indicates that company size, degree of variation, and turnover frequency have a positive relationship with financial profitability. In contrast, the degree of leverage has a negative relationship. Moreover, in the case of Italy, change in GDP also shows a positive influence on ROE. These results are consistent with those obtained by Susilo et al. (2020) for Indonesian firms in the period 2010–2017 [55], with the results of Dahmash et al. (2021) in a study conducted on Jordanian firms for the period 2011–2018 [96], and with the findings of Le et al. (2020) for Vietnamese firms in the period 2008–2015 [35].

Concerning the determinants of economic profitability, the study results indicate, similarly, that company size, growth, and turnover of assets have a positive and significant impact. In contrast, the level of indebtedness has a negative effect on ROA. The results obtained are similar to the findings of Liu et al. (2020) for Chinese companies in the agricultural sector in the period 2013–2018 [141], and to those of Altaf & Shah (2018) for Indian firms in the period 2007–2016 [142].

Both countries obtained similar results in the regression analysis for both economic and financial profitability. According to the Pecking Order Theory, a higher level of indebtedness will negatively affect profitability. Therefore, companies in the sector should reduce their dependence on borrowed funds in order to improve their income statement, as third-party financing is not free of cost.

The size of the company and its growth rate were also found to be relevant for both ROA and ROE. Companies with higher asset size and asset growth are shown to be more profitable, which supports the Resource-Based View Theory. Therefore, it would be advisable for companies, especially smaller ones, to merge, as economies of scale, negotiating power, and market strength increase financial performance.

Efficiency in the use of assets is also an important aspect to increase profitability, since asset turnover was shown to be one of the influential variables. Therefore, it is not enough to have a significant volume of assets; both working and producing assets are required to generate higher operating income and, consequently, higher results.

In the case of Italy, companies should reduce their average payment period to achieve greater economic profitability. Delays entail costs as cash payments are usually rewarded with an invoice discount and late payments are charged with interest. The difference in the payment period between the two countries is 30 days, with an average of 50.72 days for Spain and 81.02 days for Italy.

With regard to the return on equity, it is striking that the variation in GDP, i.e., the performance of the national economy, only significantly affects the profitability of Italian companies. This could be explained by the fact that Spanish companies allocate an important part of their production to exports, and are therefore less dependent on the economic situation of the country. On the other hand, Italian and Spanish companies have an average indebtedness of 61% and 44%, respectively, so external dependence is greater for the Italian companies, which are therefore more influenced by the national economy. Consequently, it would be desirable for Italian companies to focus more on exporting their production in the future.

This article provides new empirical evidence on the determinants affecting the profitability of companies belonging to the natural stone sector in Spain and Italy. It also constitutes a case study on how the productive fabrics of these countries, which both suffered the effects of the global financial crisis of 2008, had to adapt to a new economic context characterized by internationalization and competitiveness.

The findings of this empirical analysis have the following practical implications. First, companies in the natural stone sector in Spain and Italy should aim to increase in size, allowing them to take advantage of economies of scale arising from volume of operations, as well as the consequent increase in bargaining power, in order to increase profitability. Secondly, companies should seek to increase equity capital, leading to a reduction in the debt ratio, as greater financial autonomy would improve results. Thirdly, the average payment period should be reduced as much as possible, as so-called spontaneous financing is not without cost. However, this cost is not always perceived as an implicit cost. Companies could then take advantage of early payment discounts that would positively impact profitability.

We are not aware of any previous studies of this type carried out on the natural stone sector. For this reason, the present study is of considerable relevance, as it is pioneering in analyzing the profitability of companies from two countries that occupy an important place in the production and extraction of natural stone in the world.

This research is not without its limitations. It would be interesting to extend the analysis to other variables not covered in this study (management style, corporate social responsibility measures, production systems, etc.), which would require a customized survey of a large and representative number of companies. In future, the study could be extended to other countries with a developed natural stone sector to test whether the results hold or are affected by the individual characteristics of these countries. It would also be interesting to look back in a few years and study the effects of the COVID-19 crisis on the sector, i.e., study how the sector has recovered after the downturn suffered in 2020 due to the pandemic.

## Author Contributions

**Conceptualization:** María del Carmen Valls Martínez.

**Data curation:** María del Carmen Valls Martínez.

**Formal analysis:** María del Carmen Valls Martínez.

**Funding acquisition:** Fernando José Zambrano Farías.

**Investigation:** Fernando José Zambrano Farías.

**Methodology:** María del Carmen Valls Martínez.

**Software:** Fernando José Zambrano Farías.

**Supervision:** María del Carmen Valls Martínez.

**Validation:** Pedro Antonio Martín-Cervantes.

**Writing – original draft:** Fernando José Zambrano Farías, Pedro Antonio Martín-Cervantes.

**Writing – review & editing:** María del Carmen Valls Martínez.

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
