## [Decision Letter · Decision Letter 0]

9 Aug 2022

PONE-D-22-07602Profitability determinants of the natural Stone industry: evidence from Spain and ItalyPLOS ONE

Dear Dr. Zambrano Farías,

Thank you for submitting your manuscript to PLOS ONE. After careful consideration, we feel that it has merit but does not fully meet PLOS ONE’s publication criteria as it currently stands. Therefore, we invite you to submit a revised version of the manuscript that addresses the points raised during the review process.

The manuscript requires further revisions regarding the introductory section, prior literature, along with the quantitiave framework and outcomes’ discussion.

We look forward to receiving your revised manuscript.

Kind regards,

Stefan Cristian Gherghina, PhD. Habil.

Academic Editor

PLOS ONE

Journal Requirements:

Reviewers' comments:

Reviewer's Responses to Questions

**Comments to the Author**

1. Is the manuscript technically sound, and do the data support the conclusions?

Reviewer #1: Yes

Reviewer #2: Yes

2. Has the statistical analysis been performed appropriately and rigorously? 

Reviewer #1: Yes

Reviewer #2: I Don't Know

3. Have the authors made all data underlying the findings in their manuscript fully available?

Reviewer #1: Yes

Reviewer #2: Yes

4. Is the manuscript presented in an intelligible fashion and written in standard English?

Reviewer #1: Yes

Reviewer #2: Yes

5. Review Comments to the Author

Reviewer #1: The research problem undertaken by the Authors on the profitability determinants of the natural Stone industry in Spain and Italy is particularly timely and important in an economic and political context, given the shocks occurring in the EU economy due to the armed conflict in Ukraine. The article is carefully prepared, the sources used are not objectionable, up-to-date and correctly selected. The number of sites analysed is large - 453 companies (203 Spanish and 250 Italian companies) in a five-year observation time series (2015-2019). In the context of the choice of research period, however, it is worth considering extending it to at least 2020 (or even 2021), which would also capture the impact of the COVID-19 pandemic on the analysed market. The research methods used are correct, their selection having been properly justified by appropriate statistical tests. The conclusions are correct, referenced to the existing body of literature. The editorial side of the paper does not raise major objections (editorial correction of item 52 of the literature list - lines 638-640 should be made).

Reviewer #2: The topic of this article is relatively interesting but requires a major revision.

1. ABSTRACT — at the end of the passage, add the implication of your research, how it might be beneficial for company and government especially in the country where the research is conducted.

2. INTRODUCTION — need to be made more interesting, structured like an upside-down pyramid: (1) start from the general view of the research and why the object is intriguing, especially in those two countries, (2) theories surrounding the profitability, (3) existing research about profitability and how your research is filling the gap in the literature, (4) closed with the research aim and contribution.

The first paragraph also doesn’t seem to be needed, instead start with what’s interesting about the research object.

3. LITERATURE REVIEW—this section is also not structured in a good way for readers to be able to comprehend the red string of the research. This section must be filled with the theories that theories used as a basis in the research and description of all the variables, also older studies that support your hypothesis. That way conceptual framework can be built and understood clearly.

4. METHODOLOGY

a) Sample selection & data collection: elaborate more on the sampling technique and how you gathered 453 samples, and why the period 2015-2019 was selected. It’s good to be more update since this is already 2022.

b) Description of variables: in line 174 and 175, there are incorrect citing.

5. RESULTS — beside presenting the result of descriptive statistic, also tell systemically how your hypothesis is proven or not proven in each country. It is meant to answer the research question.

6. DISCUSSION — must include a discussion of every result you presented in previous section. Explain the implications of the research result for firms and countries. Lines 435-450 are not necessary in this section because it has been explained before. Focus to the discussion of result in each country. To make it more interesting, explain the contrast between those two countries and what’s your analysis regarding the difference.

7. CONCLUSION — at the end of the conclusion, add the limitation of this research and the direction of future research that you suggest.

6. PLOS authors have the option to publish the peer review history of their article (what does this mean?). If published, this will include your full peer review and any attached files.

Reviewer #1: No

Reviewer #2: No

---

## [Author Response · Author response to Decision Letter 0]

23 Sep 2022

Dear Editor and Reviewers,

We would like to thank you for taking the time to review our manuscript and for the comments you have provided, which have helped us to improve the quality of the article submitted.

We must inform you that, although we have not received any indication about improving the writing in English, and being aware of our limitations with a language that is not our mother tongue, we have sent the manuscript to a professional native translator in order to present an article in perfect English. You will be able to check the corrections made in this regard.

In addition to activating the change control, all corrections have been highlighted in red (except those referring to grammatical issues made by the translator) to facilitate the work of the reviewers.

Reviewer 1

Comment 1: The research problem undertaken by the Authors on the profitability determinants of the natural Stone industry in Spain and Italy is particularly timely and important in an economic and political context, given the shocks occurring in the EU economy due to the armed conflict in Ukraine. The article is carefully prepared, the sources used are not objectionable, up-to-date and correctly selected. The number of sites analysed is large - 453 companies (203 Spanish and 250 Italian companies) in a five-year observation time series (2015-2019).

Response: Thank you very much for your comments.

Comment 2: In the context of the choice of research period, however, it is worth considering extending it to at least 2020 (or even 2021), which would also capture the impact of the COVID-19 pandemic on the analysed market.

Response: Thank you very much for your comments.

• We have include in the Section 3.1 the following: “The year 2015 was chosen as the first year of the study because we did not want the conclusions to be outdated, considering the cycles of the economy. The study ended in 2019, as 2020 was a highly atypical year for companies, especially in Spain and Italy, two of the countries most heavily hit by the COVID-19 pandemic, with many months of paralyzed economic activity. Therefore, including the year 2020 in the analysis would have distorted the results. The period 2015-2019 represents a 5-year period of normal economic activity, thereby allowing reliable conclusions to be drawn.”.

• In Spain, companies officially present their accounts on 30 June of the following year. In other words, the accounts for 2021 were submitted to the Mercantile Register on 30 June 2022. From this point onwards, databases can start capturing information for incorporation, but the process takes several months. Therefore, the 2021 information is not yet available for this research.

• We have include in the Section 5 the following: “In future, the study could be extended to other countries with a developed natural stone sector to test whether the results hold or are affected by the individual characteristics of these countries. It would also be interesting to look back in a few years and study the effects of the COVID-19 crisis on the sector, i.e., study how the sector has recovered after the downturn suffered in 2020 due to the pandemic”.

As you know, all research can always be extended. But we researchers have to set a limit to the work we do, which does not preclude us from broadening our horizons in the future. We hope that you will be sympathetic to this aspect and consider the work meritorious in its present status.

Comment 3: The research methods used are correct, their selection having been properly justified by appropriate statistical tests. The conclusions are correct, referenced to the existing body of literature.

Response: Thank you very much for your comments.

Comment 4: The editorial side of the paper does not raise major objections (editorial correction of item 52 of the literature list - lines 638-640 should be made).

Response: Thank you very much for your comment. Has been corrected (now item 75)

Reviewer 2

Comment 1: ABSTRACT — at the end of the passage, add the implication of your research, how it might be beneficial for company and government especially in the country where the research is conducted.

Response: Thank you very much for your comment. We have followed your instructions. Please see the manuscript (highlighted in red).

Comment 2: INTRODUCTION — need to be made more interesting, structured like an upside-down pyramid: (1) start from the general view of the research and why the object is intriguing, especially in those two countries, (2) theories surrounding the profitability, (3) existing research about profitability and how your research is filling the gap in the literature, (4) closed with the research aim and contribution.

The first paragraph also doesn’t seem to be needed, instead start with what’s interesting about the research object.

Response: Thank you very much for your comments. We have deleted the first paragraph and followed your instructions. Please see the manuscript (highlighted in red).

Comment 3: LITERATURE REVIEW—this section is also not structured in a good way for readers to be able to comprehend the red string of the research. This section must be filled with the theories that theories used as a basis in the research and description of all the variables, also older studies that support your hypothesis. That way conceptual framework can be built and understood clearly.

Response: Thank you very much for your comments. We have followed your instructions. Please see the manuscript (highlighted in red). Regarding the description of the variables, this is done in Section 3.2, i.e. within Section 3, as usual in the literature.

Comment 4: METHODOLOGY

a) Sample selection & data collection: elaborate more on the sampling technique and how you gathered 453 samples, and why the period 2015-2019 was selected. It’s good to be more update since this is already 2022.

b) Description of variables: in line 174 and 175, there are incorrect citing.

Response: Thank you very much for your comments. We have corrected the citation and incorporated explanation for their comments. Please see Section 3.1 of the manuscript (highlighted in red) and the response to the Comment 2 of the Reviewer 1.

Comment 5: RESULTS — beside presenting the result of descriptive statistic, also tell systemically how your hypothesis is proven or not proven in each country. It is meant to answer the research question.

Response: Thank you very much for your comment. Please see the Section 4.2 (Multivariate analysis) of the manuscript 

Comment 6: DISCUSSION — must include a discussion of every result you presented in previous section. Explain the implications of the research result for firms and countries. Lines 435-450 are not necessary in this section because it has been explained before. Focus to the discussion of result in each country. To make it more interesting, explain the contrast between those two countries and what’s your analysis regarding the difference.

Response: Thank you very much for your comments. We have deleted the specified lines, and followed your instructions. Please see the manuscript (highlighted in red).

Comment 7: CONCLUSION — at the end of the conclusion, add the limitation of this research and the direction of future research that you suggest.

Response: Thank you very much for your comment. We have followed your instructions. Please see the manuscript (highlighted in red).

Dear Reviewers, we have done our best to fulfil all your indications. We hope that we have been able to meet your expectations and that you now consider the manuscript to have merit for publication. Thank you very much for your consideration.

Best regards,

The authors

---

## [Decision Letter · Decision Letter 1]

17 Oct 2022

Profitability determinants of the natural stone industry: evidence from Spain and Italy

PONE-D-22-07602R1

Dear Dr. Zambrano Farías,

We’re pleased to inform you that your manuscript has been judged scientifically suitable for publication and will be formally accepted for publication once it meets all outstanding technical requirements.

Kind regards,

Stefan Cristian Gherghina, PhD. Habil.

Academic Editor

PLOS ONE

Additional Editor Comments (optional):

Reviewers' comments:

Reviewer's Responses to Questions

**Comments to the Author**

1. If the authors have adequately addressed your comments raised in a previous round of review and you feel that this manuscript is now acceptable for publication, you may indicate that here to bypass the “Comments to the Author” section, enter your conflict of interest statement in the “Confidential to Editor” section, and submit your "Accept" recommendation.

Reviewer #1: All comments have been addressed

Reviewer #2: All comments have been addressed

2. Is the manuscript technically sound, and do the data support the conclusions?

Reviewer #1: Yes

Reviewer #2: Yes

3. Has the statistical analysis been performed appropriately and rigorously? 

Reviewer #1: Yes

Reviewer #2: Yes

4. Have the authors made all data underlying the findings in their manuscript fully available?

Reviewer #1: Yes

Reviewer #2: Yes

5. Is the manuscript presented in an intelligible fashion and written in standard English?

Reviewer #1: Yes

Reviewer #2: Yes

6. Review Comments to the Author

Reviewer #1: (No Response)

Reviewer #2: the author has elaborated what I suggested earlier and presented it more systematically. at the end of the abstract has been added the implications of research in each country. Yes, the data support the conslusions, the statistical analysis has been performed appropriately and rigorously and all data underlying the findings in their manuscript fully available. The author has sent the manuscript to a proffesional translator.

7. PLOS authors have the option to publish the peer review history of their article (what does this mean?). If published, this will include your full peer review and any attached files.

Reviewer #1: No

Reviewer #2: No

---

## [Editor Report · Acceptance letter]

25 Nov 2022

PONE-D-22-07602R1 

Profitability determinants of the natural stone industry: evidence from Spain and Italy 

Dear Dr. Zambrano Farías:

I'm pleased to inform you that your manuscript has been deemed suitable for publication in PLOS ONE. Congratulations! Your manuscript is now with our production department. 

Kind regards, 

on behalf of

Dr. Stefan Cristian Gherghina 

Academic Editor

PLOS ONE